# Investigating the Impact of Integration the Saudi Code of Energy Conservation with the Solar PV Systems in Residential Buildings

**Radwan A. Almasri [1,2]** , **Abdullah A. Alardhi [1,]\* and Saad Dilshad [3]**

[1]  Department of Mechanical Engineering, College of Engineering, Qassim University, Buraydah 6666-51452, Saudi Arabia; masri.radwan@qec.edu.sa or masri.radwan@gmail.com
[2]  Renewable Energy Engineering Program, College of Engineering, Qassim University, Buraydah 6666-51452, Saudi Arabia
[3]  Department of Electrical and Computer Engineering, COMSATS University Islamabad, Islamabad 45550, Pakistan; saadkhan006@gmail.com
\*  Correspondence: Aaardhi@hotmail.com

**Abstract:** The demand for air conditioning is increasing day by day in the world's hot and humid climate areas. Energy conservation in buildings can play a vital role in meeting this high cooling demand. This paper attempts to consider the impacts of energy efficiency and renewable energy measures on the energy demand of Saudi Arabia's residential buildings. The energy analysis and economic feasibility analysis of thermal insulations are performed in this paper by investigating the effect of residential buildings' thermal insulations on the economic feasibility of grid-connected photovoltaic systems. This was the combined effort of building owners and government, and buildings were examined if a photovoltaic system and thermal insulation were used. The study was conducted in the three climate zones in Saudi Arabia. The results showed that the building base case's annual electrical energy consumption in Riyadh city was 67,095 kWh, Hail 57,373 kWh, and Abha 26,799 kWh. For the basic case-building in Riyadh, 69% of the total electrical energy was used for cooling and heating. Applying the Saudi Building Code requirement for Riyadh will provide only 18% of the total energy used for cooling and heating. RETScreen 6.1 software was used to design a photovoltaic system; the analysis was done using technical and economic indicators. The annual yield factor for Riyadh, Hail, and Abha was 1649 kWh/kWp/year, 1711 kWh/kWp/year, and 1765 kWh/kWp/year, respectively. The capacity factors for Riyadh, Hail, and Abha were 18.8%, 19.5%, and 20.1%, respectively. The Unified photovoltaic Levelized energy costs were 0.031, 0.030, and 0.029 $/kWh for Riyadh, Hail, and Abha, respectively. Finally, the Net Present Value and greenhouse gas emissions reduction have been estimated.

**Keywords:** thermal insulation; residential building; SBC-602; energy efficiency; solar energy; sustainable; KSA

## 1. Introduction

The socioeconomic growth of a country is measured by energy intake per capita. Energy is key to the modern-day world, and it is vitally important for the functioning of devices. It is essential for life in any country for production and comfort, among other things, but it is necessary conserve energy and use it efficiently to promote economic sustainability. The primary energy consumption growth globally was around 1.12% in 2019 [1]. However, worldwide energy demand declined by 3.8% in the first quarter of 2020, with the vast majority of the effect felt in March as precautionary lockdown measures were authorized in a significant part of the world [2]. For extreme climates, the buildings require an active system to maintain the heating or cooling demands throughout the year. This energy consumption in buildings depends on the quality of the building's

envelope, building area, occupant behavior, and ambient climate conditions [3]. The energy consumption in the residential sector in the Kingdom of Saudi Arabia (KSA) is immense, which is against the economic and environmental goals of Saudi vision 2030 [4].

In the KSA, the oil production was around 559.3, 576.8, and 556.6 million tonnes in 2017, 2018, and 2019, respectively. This shows that a minor decrease was observed in 2019 as the world going more towards Renewable Energy (RE), and then, in 2020, COVID-19 also caused a significant halt in oil production. In energy consumption per capita, the KSA was ranked 15th globally in 2017, and it is increasing each year. This increase has also caused the KSA to reach the most $CO_2$ emissions of any year on record, reaching its peak in this decade in 2016 at 599.5 Million tonnes [1]. One reason for this continuous increase is the low cost of energy, which leads to inefficient buildings. Focusing on the task at hand, the government has introduced several strategies to control $CO_2$ emissions and simultaneously increase energy efficiency (EE) [5]. This starts by increasing the price of local fuel for industrial firms and motivating them to use energy-efficient technologies and applying a new building code called the Saudi Building Code (SBC-602) [6], which will make new buildings more energy efficient with less heating and cooling losses.

SBC-602 divides the KSA into three zones due to its different climates. As shown in Figure 1, Climate Zone 1 is the hottest and biggest zone in the KSA [6]. One reason for the high air conditioning (A/C) load is the warm and sweltering KSA climate. Among other factors, building envelopes in the KSA are made of concrete or brick without thermal insulations.

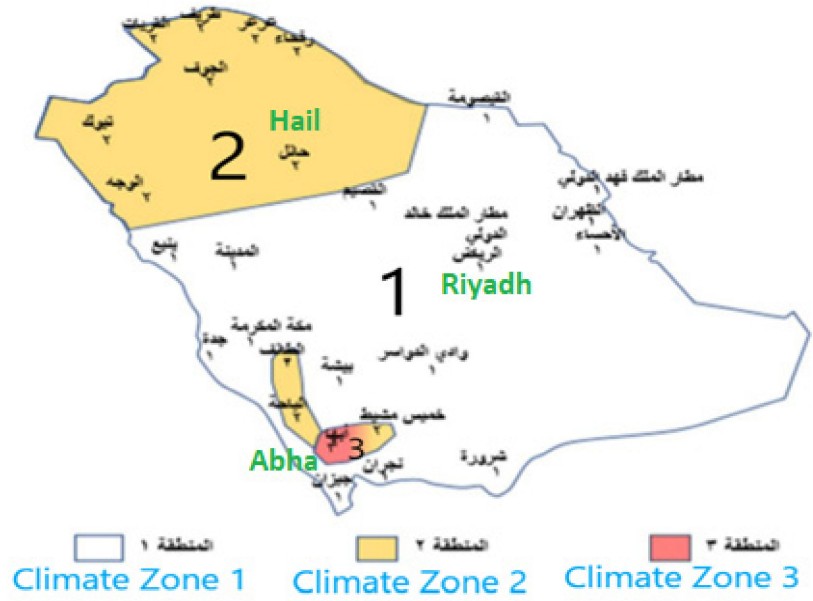

**Figure 1.** Climate zone map of the Kingdom of Saudi Arabia (KSA) [6].

The total energy demand will decrease by applying SBC-602 on the new and existing buildings with good thermal insolation, especially during peak load time. This will eliminate significant Greenhouse Gases (GHG) emissions, reduce air pollution, improve environmental conditions, and create economic development and jobs in manufacturing, installation, and more [7].

The KSA climate is harsh, and some zones are scorching hot and dry in summer, with the temperature reaching 47 °C in summer and 2 °C in winter [5]. From the data from 30 stations around KSA [8], the annual average daily global horizontal irradiation is greater on terrain and lower on the seashores, and it ranged from 5700 to 6700 Wh/m$^2$. The annual average daily direct normal irradiation was much more variable in the stations; it ranged from 4400 Wh/m$^2$ to more than 7300 Wh/m$^2$. With regard to temperatures, some locations are extremely high, and reach the annual average of over 30 °C.

A model villa investigated in this research is examined in three locations, the first location in Riyadh city (latitude 24°77′, longitude 46°73′), second is Hail (latitude 27°52′, longitude 41°69′), and the last is Abha (latitude 18°21′, longitude 42°50′). A statistical weather report of Riyadh, Hail, and Abha is presented in the Appendix A [9]. The key objective of this paper is to compare the cost of reducing the building energy needs and offsetting part of these by using PV grid-connected systems.

The key novelty of this research lies in the fact that the SBC-602 code was released in 2018. Hence, it is the first attempt to study the effect of applying the energy conservation code SBC 602 on the cost of the PV systems in the Saudi residential sector. Table 1 shows the energy utilization by different sectors in the KSA in 2017. In the Kingdom, the total energy consumption was 140.7 Mtoe, knowing that the primary source was oil products. On the other hand, these oil and gas were consumed in industry, transportation, non-energy purposes, and other purposes (such as electric power production, agriculture, and services), 33.5%, 29.7%, 20.7%, and 16.1%, respectively. Figure 2 shows the energy balance for the electricity production sector in the KSA in 2017. It is noted that 56.7% of the electricity was from natural gas, and the rest is from liquid fuel (crude oil and oil products). It is also seen that the efficiency of the general electric power generation process for the Kingdom, only 32.4%, and the rest were wasted.

**Table 1.** Energy consumption by applications in 2017 in KSA [10].

| Sectors | Energy Consumption and the Percentage by Applications | |
| --- | --- | --- |
| | **Mtoe** | **%** |
| **Industry** | 47.2 | 33.5 |
| **Transport** | 41.8 | 29.7 |
| **Other** | 22.6 | 16.1 |
| **Non-energy use** | 29.1 | 20.7 |
| **Total energy consumption by use** | 140.7 | 100 |

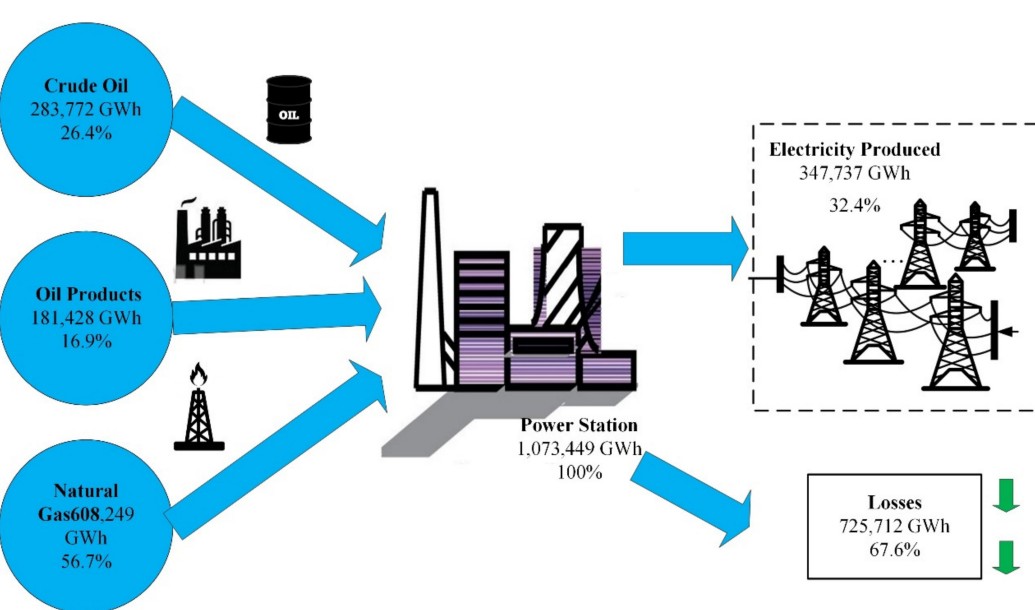

**Figure 2.** Energy balance for the electricity production sector in the KSA [10].

In 2019, the electricity sold in KSA was 288,598 GWh, which distributed as 44.5%, 16.0%, 14.1%, 19.6%, and 5.8% for residential, commercial, government, industrial, and others, respectively [11]. RE accounted for only 0.05% [12]. The electricity demand in KSA grows 5–8% annually in the last years. If this growth continues, it will reach the same



production and consumption of oil by 2035 [13]. RE resources seem to be a promising future for sustainable development for any country having fewer environmental effects [14]. The energy future in any country should be accessible and sustainable. EE and RE sources are a perfect solution to mitigate the increasing emissions and catch the rising energy demand. According to the International Renewable Energy Agency, the cost of solar and wind technologies has become lower in recent years [15].

The remainder of the paper is arranged as follows: A comprehensive literature review is presented for energy requirements in a villa, and a comparison of a grid-tied similar solar PV system is presented in Section 2. Section 3 includes the research methodology and system design description. The results and discussion and conclusion are presented in Sections 4 and 5.

## 2. Literature Review

In the literature, many studies related to energy intake in residential buildings of KSA and the Gulf Cooperation Council (GCC) and its effect on the economy and environment reviewed and their scope and challenges were highlighted. The consumption for each dwelling, per capita, and unit area was used to evaluate residential energy consumption. On the other hand, weather conditions, building characteristics, equipment owner level, and occupant behavior are the main parameters affecting this energy consumption. Balabel and Alwetaishi [16] examined different topics connected to developing the sustainable buildings sector, focusing on residential buildings in KSA. Moreover, the authors compared between suggested national rating system for buildings rating with the international LEED system. Lastly, strategies were suggested to encourage the improvement of the sustainable buildings sector. Krarti and Ihm [17] mapped the needed annual Heating Degree-Days (HDD) and annual Cooling Degree-Days (CDD) for the Middle East and North Africa region. The authors investigated the optimal design for Net-Zero Energy Building (NZEB) for prototypical single-family houses. A PV system of 2.5 to 3.0 kW was needed to accomplish the requirements of NZEB in this work. Krarti et al. [18] have used an energy analysis approach that balances EE measures economic and energy performances applying only one metric, "building energy productivity" in the GCC countries. The authors concluded that retrofitting the existing building stock can offer considerable benefits to improve building EE, like reducing fuel cost, reducing environmental damage, and creating job opportunities. Krarti and Dubey [19] evaluated the potential for retrofitting residential buildings to be energy-efficient and the ideal PV system's power to achieve the net-zero energy building requirements in Bahrain. The authors reported the benefits, reducing the annual fuel consumption needed for electricity by 62% and 55% in its peak demand compared to the real case.

There are also many studies for electrical energy consumption in the domestic sector in the KSA in literature. Ahmed et al. [20] performed a simulation study and showed that the annual energy used intensity (EUI) of a villa could be decreased from 148.8 to 72.5 kWh/m$^2$, which is 51.3% of the existing typical residential villa. The buildings profoundly depend on A/C, which uses about 80% of domestic electricity only envelope insulation. A higher Coefficient of Performance (COP) of the HVAC system is encouraging solutions for energy savings. Alaidroos and Krarti [21] have introduced a comprehensive analysis to enhance the residential building's energy performance by optimizing the envelope elements in KSA. Some efficiency measures in five different cities with different climate zone can lead to energy saving up to 39.5%. Ahmad [22] carried out a simulation study of various insulating materials used for building envelopes in Dhahran, KSA. The author reported that the cooling load distribution was 57% for walls and roofs, 22% for infiltration, 17% for windows, and 4% for others. Taleb and Sharples [23] evaluated the electrical energy usage of a residential building in Jeddah, KSA. They found that the intake could be decreased by up to 32.4% from the total by employing energy conservation measures. Aldossary et al. [24] have used a simulation study to analyze low-carbon prototype buildings' energy consumption in KSA. The authors reported a reduction of electrical energy usage to 71.6% compared

with similar houses and the consumption range of 15–34% from optimizing building characteristics. The obtained value of EUI was around 77 kWh/m$^2$ to 98 kWh/m$^2$ of electrical energy for low carbon buildings. Krarti [25] evaluated the potential and advantages of occupancy-based temperature controls in homes in KSA. This approach saved up to 38.7% of annual electricity use for a house in Riyadh and decreased up to 34.7% in peak demand in nearly all KSA region. Applying this system in all residential buildings in KSA can save up to 48 TWh/year in energy consumption, which is about 30% of the 2018 KSA residential electricity consumption. The investments here are economics with a payback of fewer than 2.0 years and suggested future practical studies in this field. Alardhi et al. [26] concluded that the annual electrical energy intake without thermal insulation in zones 1, 2, and 3 was 67,095; 57,373; and 26,799 kWh, respectively. The authors also obtained that EUI for 144.1 kWh/(m$^2$ year), 133.9 kWh/(m$^2$ year), and 128.2 kWh/(m$^2$ year) in zones 1, 2, and 3 respectively. Almushaikah and Almasri [27] show that the total annual electrical energy intake per dwelling was between 49,690 and 52,250 kWh which reflects a EUI of 131 to138 kWh/m$^2$ for the middle zone of KSA. Krarti [25] and Alaidroos and Krarti [21] pointed out that the villa's annual energy consumption lies between 139,000 and 103,000 kWh. These values are generally very high in comparison with the results of Esmaeil et al. [28] and Almasri et al. [29] who obtained the average annual electricity consumption between 29,155, and 34,448 kWh/dwelling for the same climatic condition. The discrepancy among the simulation results and the rest may be because simulations were not performed with the actual case scenarios of electricity consumption in the Saudi house, or the devices' operating time did not precisely define the A/C unit. Table 2 shows a summary of the electrical energy consumption per unit area in the residential buildings in the KSA. It is noted from Table 2 that there is a discrepancy between the values for the same type of construction, even if it is in the same climatic conditions, which necessitates the necessity of research to reach benefits that help decision-makers in the field of energy policy by taking the appropriate decision to improve the reality of energy in all technical, economic, and environmental aspects.

**Table 2.** Comparison of the annual average electricity utilized per dwelling and the energy used intensity (EUI) in the residential sector, in the KSA.

| Region or City | Climate Zone | The Annual Average Electrical Energy Consumed per Dwelling kWh/Dwelling | The Annual Average Energy Use Intensity kWh/m$^2$ | Method | Building Type/Consumer | Date of Collection Data and Reference |
|---|---|---|---|---|---|---|
| Riyadh | 1 | 103,000–139,000 | 229–309 | simulation | Villa | Krarti [25] |
| Riyadh | 1 | 119,700 | 228 | simulation | Villa | Alaidroos and Krarti [21] |
| Tobuk | 2 | 96,000 | 183 | simulation | Villa | Alaidroos and Krarti [21] |
| Abha | 3 | 67,000 | 127.6 | simulation | Villa | Alaidroos and Krarti [21] |
| Qassim | 1 | 30,031 (max 82,500) | 92.6 | electricity bills and survey | dwelling | 2012–2014 Esmaeil et al. [28] |
| The middle zone of KSA | 1 | 49,690–52,250 | 131–138 | simulation | Villa | Almushaikah and Almasri [27] |
| KSA | 1, 2, 3 | 26,799–67,095 | 128.2–144.1 | simulation | Villa | Alardhi et al. [26] |

<div align="center">Table 2. <em>Cont.</em></div>

| Region or City | Climate Zone | The Annual Average Electrical Energy Consumed per Dwelling kWh/Dwelling | The Annual Average Energy Use Intensity kWh/m$^2$ | Method | Building Type/Consumer | Date of Collection Data and Reference |
|---|---|---|---|---|---|---|
| KSA | 1, 2, 3 | 28,143 | - | statistical analysis | dwelling | 2017, [30] |
| KSA | 1, 2, 3 | - | 149.6 | statistical analysis | dwelling | 2017 Krarti et al. [31] |
| Dhahran | 1 | 52,500 | 150 | electricity bills and survey | Villa | 2012 Alrashed and Asif [32] |
| Dhahran | 1 | 35,300 | 176.5 | electricity bills and survey | dwelling | 2012 Alrashed and Asif [32] |
| Dhahran | 1 | 64,000 | 148.8 (72.5) * | simulation | Villa | Ahmed et al. [20] |
| Qassim | 1 | 29,155–34,448 | 50–60 | electricity bills and survey | dwelling | 2015–2018 Almasri et al. [29] |
| The central region of KSA | 1 | 22,000 | - | statistical analysis | consumer | 2017 [30] |
| | | 18,800–21,900 | - | statistical analysis | dwelling | 2017 and 2018 [33] |
| | | 19,000 | - | statistical analysis | consumer | 2018 [34] |

<div align="center">* Achieved by applying energy efficiency (EE) measures.</div>

Dehwah et al. [35] reported that average PV energy outputs for apartments and villas were 207 and 213 kWh/m$^2$, respectively. The total electrical energy produced in residential buildings by the PV system was 796,580 MWh, out of which 301,998 MWh generated from the apartment roof and 494,582 MWh generated from the villa roof. The authors have also listed the key factors affecting the useability of PV roofs in KSA. These factors were defined and grouped into five categories: structural, service, accessibility, shading, and other restrictions. Muhammad Asif [36] revealed that a rooftop PV system could produce around 37,746 MWh of electricity annually at King Fahd University of Petroleum and Minerals. The average energy generation was 388 kWh/m$^2$, and it saves 30.875 metric tons of $CO_2$. The average PV generation was relatively higher than other studies because it is not a residential building. Alghamdi [12] has found an 80% reduction in household electrical demand was found with the use of a proper rooftop PV. The total rooftop area is 227 m$^2$, but the utilized area excluding the area of water tanks, A/C units, etc. is 180 m$^2$. The annual electricity production by PV is 47,432 kWh. The first scenario was adapted with no Feed-In Tariff (FIT) available, and by applying net metering the obtained payback period, it is around 9.19 years, and it will be only 4.8 years via a FIT of $0.048. The estimated cost of a 28 kWp PV rooftop plant is $20,925. Imam and Al-Turki [37] reported the total annual energy generated by a grid-connected PV system installed on the rooftop of a housing building in Jeddah was 23,589 kWh, which is 86.4% of the energy demand. The minimal and actual Levelized cost of energy (LCOE) is 0.031 $/kWh and 0.038 $/kWh. The Net present value (NPV) is $4378, and simple payback is 13.8 years while the discounted payback period 14.6 years. The estimated cost of 12.25 kWp PV rooftop plant is $10,529.46, which consists of 35 mono-crystalline silicon modules, and the system result capacity factor (CF) and the performance ratio were 22% and 78%, respectively. Almarshoud [38]

has done a feasibility study of a 1 MW grid-connected PV system in the Qassim region and resulted in good energy productivity and performance indicators comparing with different countries. The yield factor, capacity factor, and Performance ratio were 2024.7, 23.1%, and 84.27%, respectively, which is high and attractive. The economic analysis of grid-connected PV systems was also performed in this research. The LCOE was less than the electricity tariff, and it was 0.0359 $/kWh, and the simple payback time (SPBT) was 13.7 years. This study gives positive results, which indicate the KSA market is a massive potential solar energy applications. Almushaikah and Almasri [27] presented the results of a recent analytical study of a PV system in the central region of KSA for a residential house. The authors reported that the average annual yield factor (YF) was between 1890.9 and 1850.5 kWh/kWp/Year and the SPBT was 13.42 years.

Akter et al. [39] have conducted economic analysis considering standalone and grid-connected PV modes having system capacities ranging from 3 to 10 kW for an Australian house. The outcomes of this research revealed that most scenarios were financially attractive by small capacity without a storage system. The payback period was more economical by a smaller size of the PV systems, and it was for a grid-connected system shorter than the standalone one. Sagani et al. [40] have presented a techno-economic and environmental evaluation of rooftop PV-grid-interconnected systems of 2.59, 4.94, 7.05, 8.93, and 9.87 kW rated power using RETScreen and SimaPro 7.1 software for Athens, Greece. On the other hand, the authors concluded that the best effect would be by small installation systems in terms of the environmental impact. The summary of similar grid-connected solar PV systems is presented in Table 3. It shows the indicators of performance and economics studied by different researchers at various places.

**Table 3.** Comparison of the performances of similar grid-connected solar PV.

| Location | System Size kWp | Annual Yield Factor kWh/kWp/Year | Annual Capacity Factor % | LCOE $/kWh | Payback Years | Reference |
|---|---|---|---|---|---|---|
| Kuwait—Al-Wafra | 100 | 1922.7 | 21.6 | 0.1 | 15 | Hajiah et al. [41] |
| Kuwait—Mutla | 100 | 1861 | 22.25 | 0.1 | 15 | Hajiah et al. [41] |
| KSA—Qassim | 1000 | 2024.7 | 23.1 | 0.036 | 13.7 | Almarshoud [38] |
| KSA—Riyadh | 11.2 | 1890.9 | 21.5 | 0.0281 | 13.4 | Almushaikah and Almasri [27] |
| Oman | - | 1696.6 | 19.46 | 0.16 | - | Kazem and Khatib [42] |
| Oman | 1000 | 1875.1 | 22.37 | 0.23 | 10 | Kazem et al. [43] |
| Meknes—Morocco | 2.04 | - | 20.20–20.52 | 0.073–0.082 | 11.1–12.69 | Allouhi et al. [44] |
| UAE—Abu Dhabi | 111.4 | 1522 | 16.5 | - | 4.7 | Emziane and Al Ali [45] |
| UAE—Abu Dhabi | 50.4 | 1802 | 20 | - | 3.9 | |
| UAE—Abu Dhabi | 215.7 | 1325 | 14 | - | - | |
| UAE—Abu Dhabi | 994 | 1438 | 16 | - | 5.2 | |
| Palestine | 5 | 1756 | - | 0.13 | 4.9 | Omar and Mahmoud [46] |

In this work, the integration of thermal insulation and solar PV was examined to show how they complement each other. What is the optimum way to utilize the benefits of both technologies?

## 3. Research Methodology

The following procedure is involved in carrying out this research:

1. The investigation was carried out through the energy analysis process using OpenStudio Software. The prices of equipment and labor are adapted from the current prices in the Saudi market.
2. The study was done for the three climate zones in KSA as per SBC-602.
3. RETscreen 6.1 software was used to design a PV system.
4. The economic feasibility was determined using the life cycle analysis method and the payback time method for both thermal insulation and PV system.
5. The environmental impact was evaluated from the energy saved and utilization of green energy for a sustainable building.

The methodology used in this work is based on changing the thermal insulation to observe the change in electrical energy consumption. It is assumed that the inside setpoint temperature to be 23 °C in the three climate zones in KSA; residential buildings will be studied in the following three scenarios:

- The building was not insulated (existing and relatively old buildings).
- It meets the construction specifications required by the Saudi Electricity Company (SEC) to connect it to the electrical grid.
- It fulfills the thermal insulation conditions in the building code SBC-602.

### 3.1. Case Study

The electricity consumption of a two-story residential building is identified as a typical case in the Kingdom. Figure 3 shows a three-dimensional block diagram of the building. Table 4 shows the specifications for the basic construction of the building. The study focused on three cities representing each climatic zone in the KSA, according to the code SBC-602, so the requirement of thermal insulation will be different, as shown in Table 5. A comparison of the three scenarios using the annual electrical energy consumption and EUI was made. A survey on a sample of homes in the Hail region was also conducted to evaluate the theoretical study results.

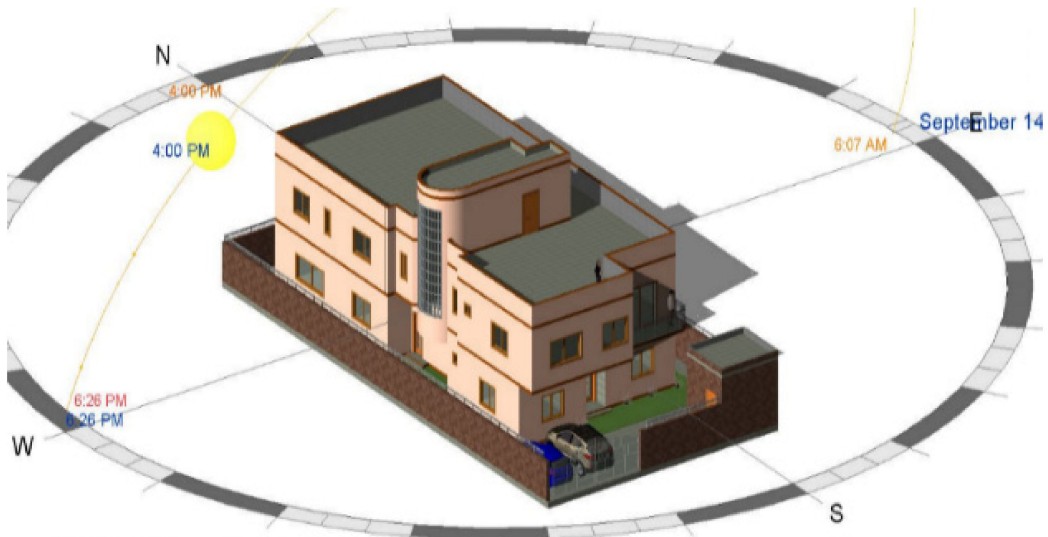

**Figure 3.** SketchUp 3-D model [26].

**Table 4.** Specific design consideration for building construction for the base case villa [26].

| | |
|---|---|
| No. of Stories | 2 |
| | U = 3.9 W/m² K and U = 2.75 W/(m² K) ( U = 2 time) |
| Total Height | 9 |
| Ground Floor area | 245 m² |
| First Floor area | 255 m² |
| The total roof area | 233 m² |
| Window area | 2.82% of the gross wall area |
| Glass type | Single-layer 5.8 W/(m² K) |
| External wall | 2 cm external Plaster + 20 cm Hollow concrete block +2 cm internal plaster, $U_{wall}$ = 3.9 W/(m² K) |
| Roof | 5 cm Tiles + 20 cm concrete roof slab + 5 cm internal plaster, $U_{roof}$ = 2.75 W/(m² K) |
| Number of occupants | 9 |

**Table 5.** The total heat transfer coefficient for walls, roof, doors, and windows [6,26].

| Thermal Insulation | Zone | U Values (W/m² K) | | | |
|---|---|---|---|---|---|
| | | Wall | Roof | Window | Door |
| **SBC-602** | Zone-1 | 0.342 | 0.202 | 2.668 | 2.839 |
| | Zone-2 | 0.397 | 0.238 | 2.668 | 2.839 |
| | Zone-3 | 0.453 | 0.273 | 2.668 | 2.839 |
| **SEC** | All KSA | 1.75 | 0.6 | 2.9 | 5 |
| **Base case** | | 3.9 | 2.75 | 5.8 | 5 |

### 3.2. General Requirements for Connecting a Small-Scale PV System

The board of directors of the Electricity and Co-Generation Regulatory Authority (ECRA) approved the amendment of the regulatory framework for small solar PV systems (PV size >1 kW and <2 MW) in the KSA [47]. This regulatory framework adopts net billing. SEC has also set some rules and regulations and a general requirement for connecting a grid-connected PV system. Figure 4 shows the requirement of protection and control devices for a small-scale PV system in descending priority order from 1 to 8 [48]. ECRA has identified the financial compensation rates for small solar PV systems services, and the FIT for surplus energy is presented in Tables 6 and 7.

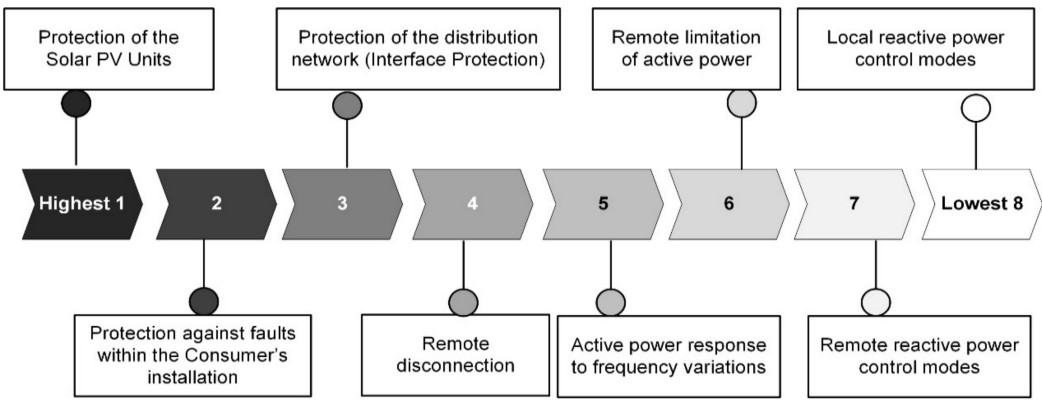

**Figure 4.** The protection and control devices of a PV System in priority ranking (from highest to lowest) [47].

**Table 6.** Feed-In Tariff (FIT) for surplus energy per sector [47].

| Sector | FI ($/kWh) |
|---|---|
| Residential sector | 0.019 |
| Other Sectors | Determined by ECRA |

**Table 7.** Services Cost by SEC [47].

| Stage | System Size (kW) | Cost ($) |
|---|---|---|
| Initial inquiry | ≤50 | 40 |
| | ≥50 | 133 |
| Connection | ≤50 | 147 |
| | ≥50 | 480 |
| Total | ≤50 | 187 |
| | ≥50 | 613 |

Services cost by SEC was presented. Although strict rules and regulations were applied for installing a grid-tied PV system, some other issues remain in a grid-connected PV system like low inertial, current harmonics, intermittency, SSR in PV parks, etc. [49]. However, mitigation of these is not in the scope of this research.

### 3.3. System Analysis

Canadian Solar modules (Type: HiKu CS3W-400P) were selected due to their high module efficiency reaching 18.11% at standard test conditions (STC). Moreover, these modules are available in the local market and are suitable for the very dusty and harsh climate of KSA. The module has a nominal max power of 400 Wp under STC. The parameters of the designed PV system are shown in Table 8.

**Table 8.** Parameter of designed Rooftop PV system.

| Total Roof Area (m$^2$) | Module Area (m$^2$) | NO# of Panels | PV System Area (m$^2$) | GRC (%) | System Capacity (kWp) |
|---|---|---|---|---|---|
| 233 | 2.2 | 46 | 101 | 43.3 | 18.4 |

For scenarios in all locations, the unified PV system is the maximum possible per the available area, considering the structural restrictions, service restrictions, accessibility, and shading restrictions. As shown in the 3D layout in Figure 5. The slope angles are 25 degrees for all three cities. The maximum number of modules can be installed 46 modules. This system needs three inverters, from Fronius company with 5 kW$_{AC}$ (7.5 kW$_{DC}$) maximum power.

The RETScreen 6.1 software [50] was used to design the unified PV system. From RETScreen data, the weather condition data for all locations is shown in the Appendix A [9].

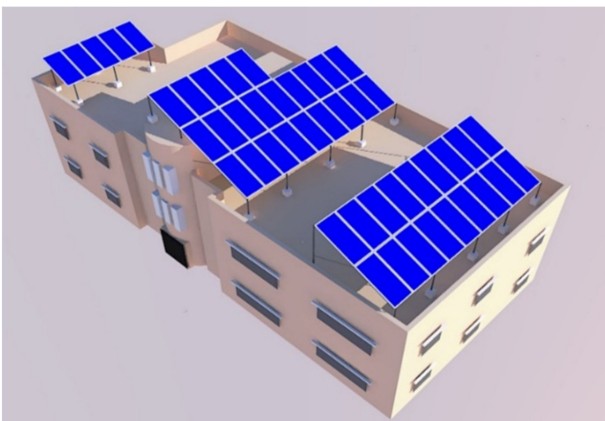

**Figure 5.** Unified PV system 3D layout.

*3.4. Performance Indicators*

Several performance indicators are frequently used to evaluate the performance of the grid-connected PV system, which are mainly yield factor (YF) and the Capacity Factor (CF) [38]; the YF is obtained by Equation (1):

$$YF = \frac{\text{Energy Produced } (kWh/year)}{\text{PV array } (kW/peak\ at\ STC)} \tag{1}$$

The CF calculates the percentage of usability of the PV array, and it is obtained by Equation (2) [38]:

$$CF = \frac{YF}{8760} \tag{2}$$

Here, 8760 refers to the total number of hours in the year.

*3.5. Financial Indicators*

Several financial indicators are usually used to assess the economic viability of an energy project. Some of these indicators are described below:

Life Cycle Cost (LCC) is the total costs of the system over the project life in today's money (by using present worth for any future money) [38]. It is obtained by Equation (3):

$$LCC = C\ \text{capital} + \sum C_{O\&M} \sum C_{replacement} - C_{salvages} \tag{3}$$

where, $C_{capital}$, $C_{O\&M}$, $C_{replacement}$, and $C_{salvage}$ are basically for the capital, operation, and maintenance, replacement, and salvage value cost. Assigning a $C_{Salvage}$ is common practice to recover a value of 30% of the original price for mechanical apparatus that can be moved.

Levelized cost of energy or Levelized cost of saved energy (LCOE/LCSE) is the average cost of energy saved from thermal insulation or that produced by PV system ($/kWh) over the project life given by Almarshoud [38]. LCSE formula for thermal insulation:

$$LCSE = \frac{C_c}{\sum E_{saved}(during\ project\ life)} \tag{4}$$

LCOE formula for PV system:

$$LCOE = \frac{LCC}{\sum E_{grid}(during\ project\ life\ with\ 1\%\ \text{deg}radation)} \tag{5}$$

The simple payback time (SPBT) is the length of time it takes to recoup the project's initial cost [38]. The SPBT is obtained by Equations (6) and (7):

The SPBT formula for thermal insulation:

$$SPBT = \frac{Initial\ \cos t}{Saved\ energy\ (kWh/yr) \times 0.048\ (USD/kWh)} \tag{6}$$

Here, 0.048 $/kWh is the cost of one kWh in Saudi Arabia.
The SPBT formula for PV system:

$$SPBT = \frac{Initial\ \cos t}{Surplus\ energy\ (kWh/yr) \times 0.019 + E\ used\ Energy\ (kWh/yr) \times 0.048 - C_{O\&M}} \tag{7}$$

Here, 0.019 and 0.048 are the FIT and cost of one kWh in KSA.

Not discounting payback time (NDPBT) is another most useful indicator of the time needed for the accumulative fuel savings to equal the total initial investments. This time obtained without discounting the fuel savings [51]. The NDPBT is obtained by Equations (8) and (9):

The NDPBT formula for thermal insulation:

$$NDPBT = \frac{\ln\left(\frac{Cs \times if}{E\ saved \times 0.048} + 1\right)}{\ln(1 + if)} \tag{8}$$

The NDPBT formula for PV:

$$NDPBT = \frac{\ln\left(\frac{LCC \times if}{Surplus\ Energy \times 0.019 + E\ used \times 0.048} + 1\right)}{\ln(1 + if)} \tag{9}$$

Net present value (NPV) [51] is obtained by the Equation (10):

$$NPV = \sum\left(\frac{Revenue\ or\ saving\ (n) \times (1 + if)^{n-1} - \cos t(n)}{(1 + d)^n}\right) \tag{10}$$

### 3.6. Materials Price Estimation of the Thermal Insulation

A cost analysis was done based on the KSA local market for supplying and installing the thermal insulation requirements of SEC and SBC-602. For walls, cost data were collected for different thermal insulation systems; this data includes the cost per meter square of wall or roof, which is the incremental cost from the base case walls (base case walls has 20 cm Hollow Concrete Blocks (HCB)). The used insulation system is called the exterior insulation finishing system (EIFS); see Figure 6. For the roof insulation, a 5 cm polystyrene was added for SEC requirements and 10 cm for SBC-602 requirements. The window insulation for both requirements was double glazed. Table 9 contains the characteristics of each building element. Its insulation system also details the increment of unit price and the area of each building element based on the building area data in Table 10.

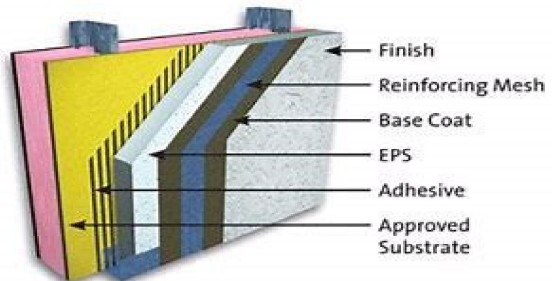

**Figure 6.** Exterior insulation finishing system (EIFS) components.

**Table 9.** The capital cost estimation of the requirements of SEC and SBC-602.

| Thermal Insulation Requirements | Details of the System | U Values (W/m² K) | Unit | QTY | Increment Unit Price $ | Sub-Total $ | Source |
|---|---|---|---|---|---|---|---|
| SEC | 20 cm insulated Hollow concrete block + Two side Plaster (walls) | 1.75 | m² | 627 | 4 | 2508 | Local Suppliers |
| | 5 cm polystyrene with the concrete (Roof) | 0.6 | m² | 233 | 7 | 1631 | Local Suppliers |
| | Double Glazed (Windows) | 2.668 | m² | 51 | 47 | 2397 | Local Suppliers |
| | Total of SEC ($) | | | | | 6536 | |
| SBC-602 | 20 HCB + 8 CM EIFS + One Side plaster (walls) | 0.31 | m² | 627 | 23 | 14,421 | Jotun Paints |
| | 10 cm polystyrene with the concrete (Roof) | 0.2 | m² | 233 | 11 | 2563 | Local Suppliers |
| | Double Glazed (Windows) | 2.668 | m² | 51 | 47 | 2397 | Local Suppliers |
| | Total of SBC-602 ($) | | | | | 19,381 | |

**Table 10.** Area of walls and windows of the building.

| Side | Walls Area (m²) | Windows Area (m²) |
|---|---|---|
| East | 208 | 14 |
| South | 115 | 10 |
| West | 188 | 15 |
| North | 120 | 12 |

*3.7. Materials Price Estimation of the PV System*

Materials price estimation was also required for the PV system. Table 11 shows the overall materials needed to install the unified PV system. For the unified PV system, the structure cost was 72 $/kW, and the inverter was 278 $/kW, other electrical components and balance of system (BOS) was 94 $/kW. All the PV scenarios cost estimation was shown in Table 12. The inflation rate was assumed as 2% as the Saudi Arabian Monetary Authority report; also, the fuel cost escalation rate is 2%, and the discount rate was supposed to be 4.5% [52]. The life of the PV panels for 25 years.

**Table 11.** The cost estimation for the unified PV systems components.

| Item | Unit | QTY. | Unit Price $ | Total Price $ |
|---|---|---|---|---|
| PV module Type: Canadian Solar HiKu CS3W-400P | No. | 46 | 120 | 5520 |
| The mounting structure for 46 Modules including concrete blocks | L.S. | 1 | 1333 | 1333 |
| Grid-Inverter Type: Fronius Primo 5.0-1 | No. | 3 | 1707 | 5121 |
| Double pole DC Fuse holder with fuse rating 20/32 A 1000 Vdc | No. | 6 | 18 | 108 |
| DC surge protection device (SPD) Type 2, 1000 Vdc/40 KA | No. | 6 | 27 | 162 |
| Miniature circuit breaker 1 × 25 A | No. | 3 | 5 | 15 |
| Miniature circuit breaker 2 × 63 A | No. | 1 | 27 | 27 |
| AC SPD Type-2, 40 KA | No. | 1 | 27 | 27 |
| IP65 External Galvanized steel box size 30 × 40 cm | No. | 3 | 53 | 162 |
| AC Panel size 40 × 60, including busbars, ducts, terminals, wires . . . etc. | No. | 1 | 179 | 179 |
| DC wire 0.9/1.8 KV$_{DC}$ 6 sq.mm | M | 250 | 1.3 | 325 |
| PVC conduit 2 inch | M | 25 | 1.3 | 32.5 |
| XLPE cable 0.6/1 KV$_{AC}$ 3 × 6 sq.mm | M | 5 | 1.3 | 6.5 |
| XLPE cable 0.6/1 KV$_{AC}$ 3 × 16 sq.mm | M | 20 | 2.7 | 58 |
| MC4 connectors (male and female) | No. | 8 | 7 | 56 |
| Earthling system | L.S. | 1 | 180 | 180 |
| Installation | L.S. | 1 | 394 | 394 |
| **Total Price of PV (capital Cost) $** | | | | **13,707** |

**Table 12.** Cost Estimation for all the PV size scenarios using room temperature (23 °C).

| City | Thermal Insulation | Load Coverage Percentage % | NO# of Panels | System Capacity (kWp) | Panels Cost $ | Structure Cost $ | Inverter Cost $ | Electrical & BOS $ | Capital Cost $ |
|---|---|---|---|---|---|---|---|---|---|
| Riyadh | Base case | 45.2 | 46 | 18.4 | 5520 | 1333 | 5120 | 1733 | 13,707 |
| | | 40.3 | 41 | 16.4 | 4920 | 1185 | 4561 | 1544 | 12,210 |
| | SEC | 77.0 | 46 | 18.4 | 5520 | 1333 | 5120 | 1733 | 13,707 |
| | | 60.3 | 36 | 14.4 | 4320 | 1041 | 4005 | 1356 | 10,721 |
| | | 40.2 | 24 | 9.6 | 2880 | 694 | 2670 | 904 | 7148 |
| | SBC-602 | 134.0 | 46 | 18.4 | 5520 | 1333 | 5120 | 1733 | 13,707 |
| | | 99.0 | 34 | 13.6 | 4080 | 983 | 3783 | 1280 | 10,126 |
| | | 81.5 | 28 | 11.2 | 3360 | 809 | 3115 | 1054 | 8339 |
| | | 61.2 | 21 | 8.4 | 2520 | 607 | 2336 | 791 | 6254 |
| | | 40.8 | 14 | 5.6 | 1680 | 405 | 1558 | 527 | 4169 |
| Hail | Base case | 54.9 | 46 | 18.4 | 5520 | 1333 | 5120 | 1733 | 13,707 |
| | | 40.6 | 34 | 13.6 | 4080 | 983 | 3783 | 1280 | 10,126 |
| | SEC | 93.0 | 46 | 18.4 | 5520 | 1333 | 5120 | 1733 | 13,707 |
| | | 80.8 | 40 | 16 | 4800 | 1156 | 4450 | 1506 | 11,913 |
| | | 60.6 | 30 | 12 | 3600 | 867 | 3338 | 1130 | 8934 |
| | | 40.4 | 20 | 8 | 2400 | 578 | 2225 | 753 | 5956 |

| City | Thermal Insulation | Load Coverage Percentage % | NO# of Panels | System Capacity (kWp) | Panels Cost $ | Structure Cost $ | Inverter Cost $ | Electrical & BOS $ | Capital Cost $ |
|------|------|------|------|------|------|------|------|------|------|
|  |  | 137.8 | 46 | 18.4 | 5520 | 1333 | 5120 | 1733 | 13,707 |
|  |  | 98.8 | 33 | 13.2 | 3960 | 954 | 3671 | 1243 | 9828 |
|  | SBC-602 | 80.9 | 27 | 10.8 | 3240 | 780 | 3004 | 1017 | 8041 |
|  |  | 59.9 | 20 | 8.5 | 2400 | 614 | 2364 | 800 | 6179 |
|  |  | 41.9 | 14 | 5.6 | 1680 | 405 | 1558 | 527 | 4169 |
|  |  | 121.2 | 46 | 18.4 | 5520 | 1333 | 5120 | 1733 | 13,707 |
|  |  | 97.5 | 37 | 14.8 | 4440 | 1070 | 4116 | 1393 | 11,019 |
|  | Base case | 79.0 | 30 | 12 | 3600 | 867 | 3338 | 1130 | 8934 |
|  |  | 60.6 | 23 | 9.2 | 2760 | 665 | 2559 | 866 | 6850 |
|  |  | 39.5 | 15 | 6 | 1800 | 434 | 1669 | 565 | 4467 |
|  |  | 161.4 | 46 | 18.4 | 5520 | 1333 | 5120 | 1733 | 13,707 |
|  |  | 98.3 | 28 | 11.2 | 3360 | 809 | 3115 | 1054 | 8339 |
| Abha | SEC | 80.7 | 23 | 9.2 | 2760 | 665 | 2559 | 866 | 6850 |
|  |  | 59.7 | 17 | 6.8 | 2040 | 491 | 1891 | 640 | 5063 |
|  |  | 52.6 | 11 | 4.4 | 1320 | 318 | 1224 | 414 | 3276 |
|  |  | 172.1 | 46 | 18.4 | 5520 | 1333 | 5120 | 1733 | 13,707 |
|  |  | 97.3 | 26 | 10.4 | 3120 | 752 | 2893 | 979 | 7743 |
|  | SBC-602 | 78.6 | 21 | 8.4 | 2520 | 607 | 2336 | 791 | 6254 |
|  |  | 59.9 | 16 | 6.4 | 1920 | 463 | 1780 | 602 | 4765 |
|  |  | 37.4 | 10 | 4 | 1200 | 289 | 1113 | 377 | 2978 |

## 4. Results and Discussion

In this section, the results for electrical energy consumption, end-use energy distribution, the energy generated by different scenarios of PV systems will be determined, technical, economic, and environmental indicators analyzed.

### 4.1. Electrical Energy Consumption

The proposed residential building's monthly electrical energy consumption curve in the base case in the three cities with a setpoint temperature of 23 °C is shown in Figure 7. It is noted from the figure that the consumption of electrical energy in summer was high in Riyadh and Hail. In Abha, the energy need was stable throughout the year. This shows the importance of applying thermal insulation code SBC-602 in Riyadh and Hail. It is clear from Figure 8 that there is a big difference in annual electrical energy consumption from zone to zone and from thermal insulation scenarios to others. The annual electrical energy consumption of the building base case was 67,095; 57,373; and 26,799 kWh in Riyadh, Hail, and Abha, respectively [26]. For Riyadh, where the EUI value ranged from 100 to 162 kWh/(m$^2$ year), it is found that the results fall within the range that was obtained for many researchers previously. Except for the results by Alaidroos and Krarti [21], which are relatively high. A reduction of electricity consumption when applying thermal insulation SBC-602 ranging from 63.9% to 72.2% for Riyadh, 58.1% to 67.3% for Hail, and 32.7% to 49.1% for Abha.

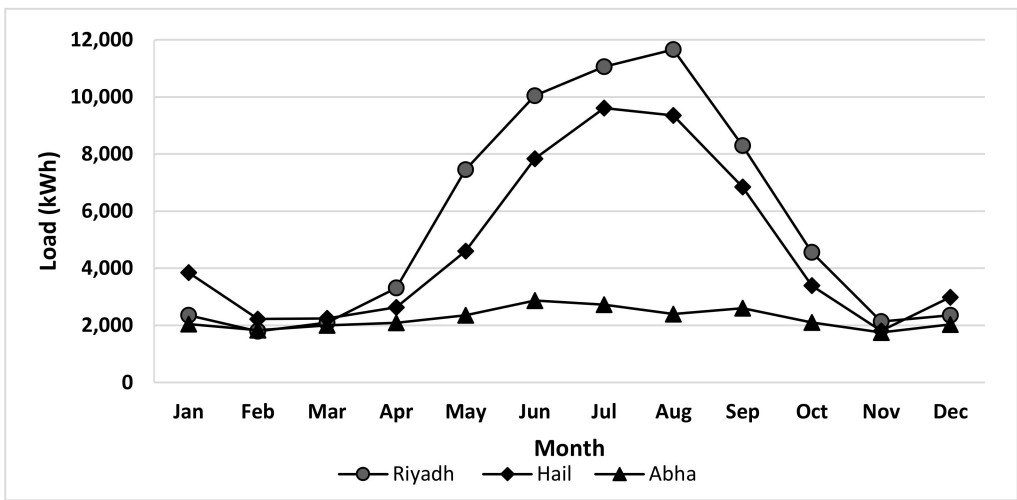

**Figure 7.** Base case monthly electrical energy consumption in the three cities at 23 °C.

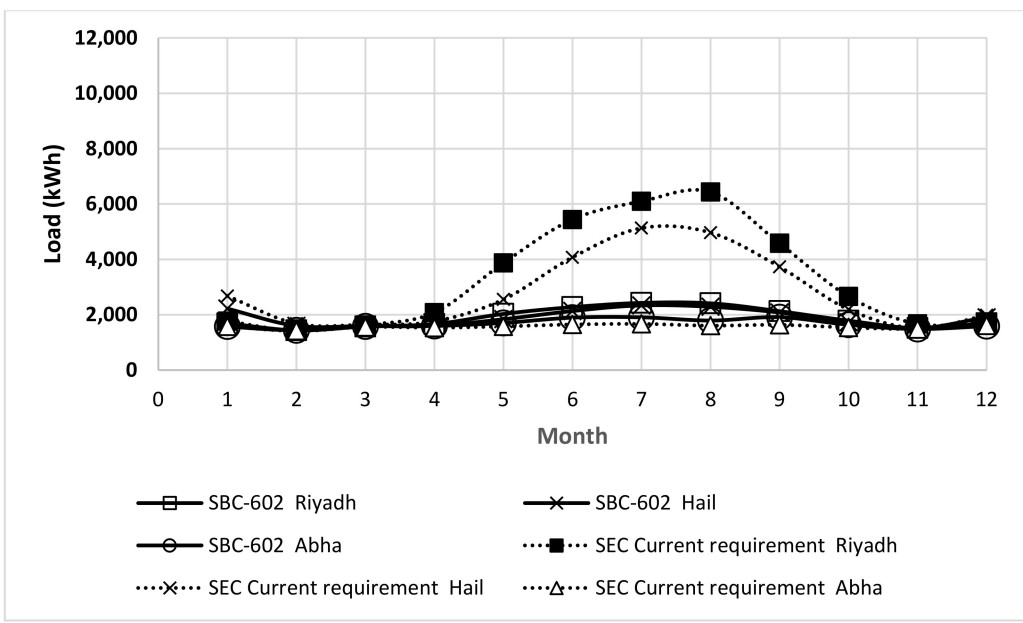

**Figure 8.** Compare the total monthly electrical energy consumptions for Saudi Electricity Company (SEC) and Saudi Building Code (SBC-602) requirements at 23 °C.

A survey was conducted on a sample of homes in the Hail region to evaluate the obtained simulation results. Dwellings were investigated from January to December 2019, based on their monthly electricity bills, the electrical energy consumption in residential buildings, the building area, and the number of occupants. The EUI and the electrical energy consumption per capita were calculated. The average annual EUI of the building without thermal insulation was 109.1 kWh/m$^2$, and for the building adapting the SEC requirement, the average was 64.9 kWh/m$^2$. A comparison between the modeling results of the basic case building or a state in which the SEC requirement was applied for Hail shows that the survey values are at the minimum level compared to the modeling results; the reason may be due to the high electricity tariffs for the residential sector which were raised at the beginning of 2018. A comparison is presented for annual electrical energy end-use distribution for the three selected cities for the base case, SEC and SEC-602, at 23 °C. For Riyadh city, 69% of the total energy is for cooling and heating, and it has been noticed that if the current requirement of SEC is applied, then it will decrease to 51%, and

it will be 18% when the SBC-602 requirement was implemented; see Table 13 for other investigated locations.

**Table 13.** Percentage of annual electrical energy end-use for cooling and heating by indoor temperature 23 °C.

|  | **Base Case** | **SEC** | **SBC-602** |
|---|---|---|---|
| **Riyadh** | 69% | 51% | 18% |
| **Hail** | 64% | 43% | 19% |
| **Abha** | 30% | 8% | 3% |

### 4.2. PV Energy System

The RETScreen simulation resulting, the total energy injected into the grid for Riyadh, Hail, and Abha is 30,348; 31,485; and 32,476 kWh/year, respectively. For the RESTScreen monthly electrical energy production results, see Figure 9.

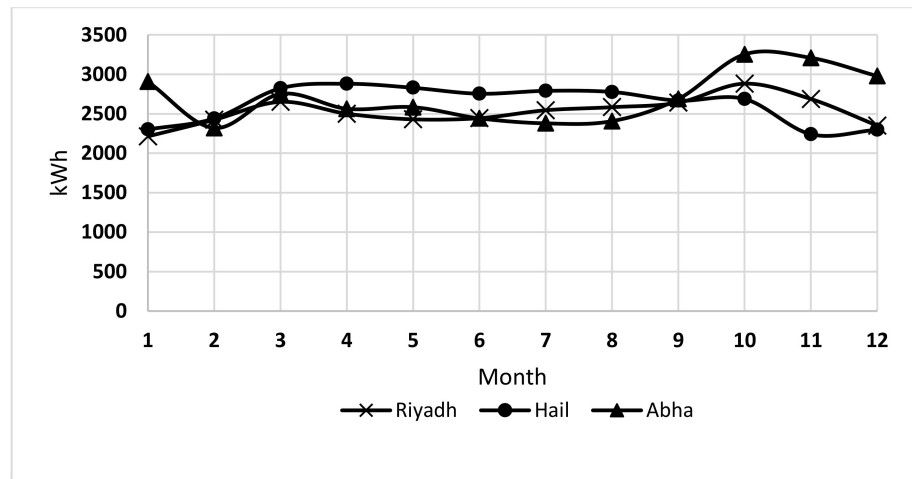

**Figure 9.** PV monthly electrical energy production.

In the unified PV system size, the designed rooftop PV system is the broadest possible, covering a certain different percentage of the needed energy based on the scenarios load; as shown in Table 14, these are for the unified PV system in each loading scenario. In this section, a PV system is designed in RETScreen software to cover less than the loading scenario's need for energy. For the base case, the maximum coverage percentage of the need load is 45.2%, 54.9%, and 121.2% for Riyadh, Hail, and Abha, respectively. For Riyadh and Hail base case, the PV system size cannot increase more, since no more available area on the roof. However, in Abha, base case needs to consider another PV scenario less than the unified PV size since its energy surplus is too high. Table 14 shows all the other PV size scenario details. Table 14 shows the building energy needs based on the different thermal insulation scenarios and how much the coverage percentage of PV energy is in relation to the need for energy. It also shows the ground coverage ratio (GRC), which is the coverage percentage of the PV system area to the roof area.

**Table 14.** PV system scenarios.

| City | Load Scenario | Building Energy Need kWh | PV Energy kWh | Load Coverage Percentage % | NO# of Panels | System Capacity (kWp) | PV System Area m$^2$ | GRC % |
|---|---|---|---|---|---|---|---|---|
| Riyadh | Base case (23 °C) | 67,095 | 30,348 | 45.2 | 46 | 18.4 | 101.2 | 43.4 |
| | | | 27,049 | 40.3 | 41 | 16.4 | 90.2 | 38.7 |
| | SEC (23 °C) | 39,390 | 30,348 | 77.0 | 46 | 18.4 | 101.2 | 43.4 |
| | | | 23,750 | 60.3 | 36 | 14.4 | 79.2 | 34.0 |
| | | | 15,833 | 40.2 | 24 | 9.6 | 52.8 | 22.7 |
| | SBC -602 (23 °C) | 22,654 | 30,348 | 134.0 | 46 | 18.4 | 101.2 | 43.4 |
| | | | 22,430 | 99.0 | 34 | 13.6 | 74.8 | 32.1 |
| | | | 18,472 | 81.5 | 28 | 11.2 | 61.6 | 26.4 |
| | | | 13,854 | 61.2 | 21 | 8.4 | 46.2 | 19.8 |
| | | | 9236 | 40.8 | 14 | 5.6 | 30.8 | 13.2 |
| Hail | Base case (23 °C) | 57,373 | 31,485 | 54.9 | 46 | 18.4 | 101.2 | 43.4 |
| | | | 23,272 | 40.6 | 34 | 13.6 | 74.8 | 32.1 |
| | SEC (23 °C) | 33,907 | 31,485 | 54.9 | 46 | 18.4 | 101.2 | 43.4 |
| | | | 27,379 | 80.8 | 40 | 16 | 88 | 37.8 |
| | | | 20,534 | 60.6 | 30 | 12 | 66 | 28.3 |
| | | | 13,689 | 40.4 | 20 | 8 | 44 | 18.9 |
| | SBC -602 (23 °C) | 22,854 | 31,485 | 137.8 | 46 | 18.4 | 101.2 | 43.4 |
| | | | 22,587 | 98.8 | 33 | 13.2 | 72.6 | 31.2 |
| | | | 18,481 | 80.9 | 27 | 10.8 | 59.4 | 25.5 |
| | | | 13,689 | 59.9 | 20 | 8.5 | 44 | 18.9 |
| | | | 9583 | 41.9 | 14 | 5.6 | 30.8 | 13.2 |
| Abha | Base case (23 °C) | 26,799 | 32,476 | 121.2 | 46 | 18.4 | 101.2 | 43.4 |
| | | | 26,122 | 97.5 | 37 | 14.8 | 81.4 | 34.9 |
| | | | 21,180 | 79.0 | 30 | 12 | 66 | 28.3 |
| | | | 16,238 | 60.6 | 23 | 9.2 | 50.6 | 21.7 |
| | | | 10,590 | 39.5 | 15 | 6 | 33 | 14.2 |
| | SEC (23 °C) | 20,120 | 32,476 | 161.4 | 46 | 18.4 | 101.2 | 43.4 |
| | | | 19,768 | 98.3 | 28 | 11.2 | 61.6 | 26.4 |
| | | | 16,238 | 80.7 | 23 | 9.2 | 50.6 | 21.7 |
| | | | 12,002 | 59.7 | 17 | 6.8 | 37.4 | 16.1 |
| | | | 10,590 | 52.6 | 11 | 4.4 | 24.2 | 10.4 |
| | SBC -602 (23 °C) | 18,874 | 32,476 | 172.1 | 46 | 18.4 | 101.2 | 43.4 |
| | | | 18,356 | 97.3 | 26 | 10.4 | 57.2 | 24.6 |
| | | | 14,826 | 78.6 | 21 | 8.4 | 46.2 | 19.8 |
| | | | 11,296 | 59.9 | 16 | 6.4 | 35.2 | 15.1 |
| | | | 7060 | 37.4 | 10 | 4 | 22 | 9.4 |

### 4.3. Performance Indicators

Table 15 shows the annual YF was 1649, 1711, and 1765 kWh/kWp/year for Riyadh, Hail, and Abha, respectively. These values of YF relatively high compared to similar grid-connected PV systems, as shown in Table 3. Abha has the best YF due to the cool weather, which causes better efficiency of the module and leads to the best energy production. The second important technical performance indicator is CF. The annual CF is 18.8%, 19.5%, and 20.1% for Riyadh, Hail, and Abha, respectively. The obtained CF for three different sites at KSA was as attractive as the rest of the middle east, as shown in Table 3.

**Table 15.** PV Technical performance indicators.

|  | **Riyadh** | **Hail** | **Abha** |
|---|---|---|---|
| **Annual Yield factor (kWh/kWp/year)** | 1649 | 1711 | 1765 |
| **Annual Capacity factor (CF) (%)** | 18.8 | 19.5 | 20.1 |

Table 16 shows the monthly average of the unified PV system production for the three locations. For example, in January, it was noticed Hail PV production was 2301 kWh. In Riyadh, it was 2213 kWh, which is less than Hail even when Riyadh in the same month has higher solar irradiation; this may be due to the ambient temperature effect on the PV.

**Table 16.** RETScreen average unified PV energy production (kWh) for the three locations.

| City | System Capacity kWp | Jan | Feb | Mar | Apr | May | Jun | Jul | Aug | Sep | Oct | Nov | Dec | Average |
|---|---|---|---|---|---|---|---|---|---|---|---|---|---|---|
| Riyadh |  | 2213 | 2423 | 2651 | 2501 | 2430 | 2444 | 2544 | 2584 | 2641 | 2881 | 2686 | 2349 | 2529 |
| Hail | 18.4 | 2301 | 2439 | 2823 | 2880 | 2830 | 2754 | 2791 | 2776 | 2667 | 2687 | 2240 | 2297 | 2624 |
| Abha |  | 2906 | 2319 | 2752 | 2563 | 2583 | 2442 | 2379 | 2406 | 2687 | 3253 | 3208 | 2977 | 2706 |

When the achieved results for the performance indicator were correlated with the existing studies in literature like Almarshoud [38], almost similar results are observed with a slight difference in the simulations. The research involves a 1 MW grid-connected PV system in the Qassim region. From the selected three cities in this study, Riyadh and Qassim are from climate zone 1, as climate zone 1 is the hottest zone in KSA. The annual YF is 1756 kWh/kWp/year, whereas, for the Qassim, it was reported as 2024.7 kWh/kWp/year for a 1 MW grid-connected PV system [38]. Similarly, the Annual CF is 20.1%, and it was reported for the same climate zone 23.1% [38]. The difference may be a relatively higher temperature during the winter season in the Qassim region than in Riyadh. Some more similar results were also reported by Hajiah et al. [41]. The authors studied a 100 kWp system for the cities of Kuwait-Al-Wafra and Kuwait–Mutla with reported annual YF of 1922.7 and 1861 kWh/kWp/year, respectively. The results obtained in this study are very closer to the Mutla in Kuwait. Similarly, the CF is reported to 21.6% and 22.25% for Kuwait-Al-Wafra and Kuwait–Mutla, respectively.

### 4.4. Saved and Generated Energy

The main objective of this work is to make an economic comparison between the cost of reducing the need for energy and offsetting part of the energy needs of the building by using PV grid-connected systems. Table 17 illustrates the saved energy from implementing the SEC's current requirements and the implementation of energy efficiency measures in SBC-602 in the three cities. Moreover, it shows the generated energy by the unified PV systems for each city.

**Table 17.** Annual saved and generated energy at 23 °C.

| Thermal Insulation | Riyadh (Zone 1) | | | Hail (Zone 2) | | | Abha (Zone 3) | | |
| | Saved Energy kWh/Year | Unified PV Energy kWh/Year | En kWh/Year | Saved Energy kWh/Year | Unified PV Energy kWh/Year | En kWh/Year | Saved Energy kWh/Year | Unified PV Energy kWh/Year |
|---|---|---|---|---|---|---|---|---|---|
| Base Case | 0 | | 57,373 | 0 | | 26,799 | 0 | |
| SEC | 27,705 | 30,348 | 33,907 | 23,466 | 31,485 | 20,120 | 6679 | 32,476 |
| SBC-602 | 43,283 | | 22,854 | 34,519 | | 18,874 | 7925 | |

The ECRA framework for small solar PV systems says the FIT for surplus energy is 0.019 $/kWh. Several scenarios for different cities were taken to observe their requirements. In some scenarios, the energy need of the building is more than PV-produced energy. In this scenario, all the PV-produced energy will consider its FIT is 0.048 $/kWh since it saves the owner from paying $0.048 for the used kilowatt-hour. Another scenario when the PV produced energy was more than the building energy needs to have a different FIT. The first one is 0.048 $/kWh for the building energy need, and the second is for surplus energy from the PV, and its FIT is 0.019 $/kWh. The ECRA framework does not show the payment method; in this work, it is considered an annual payment. For more details about the PV saving for these scenarios, the first-year saving is shown in Table 18. Table 19 presents the first year saving by applying thermal insulation.

**Table 18.** PV First-year saving based on RETScreen results at 23 °C.

| City | Thermal Insulation | PV Production | Surplus Energy | Used Energy (0.048 $/kWh) | | FIT 0.019 $/kWh | | First-Year Saving $ |
| | | | | Energy kWh | Money-Saving $ | Energy kWh | Money-Saving $ | |
|---|---|---|---|---|---|---|---|---|
| Riyadh | Base case | 30,348 | 0 | 30,348 | 1457 | 0 | 0 | 1457 |
| | SEC | | 0 | 30,348 | 1457 | 0 | 0 | 1457 |
| | SBC-602 | | 6535 | 23,812 | 1143 | 6535 | 124 | 1267 |
| Hail | Base case | 31,485 | 0 | 31,485 | 1511 | 0 | 0 | 1511 |
| | SEC | | 0 | 31,485 | 1511 | 0 | 0 | 1511 |
| | SBC-602 | | 8631 | 22,854 | 1097 | 8631 | 164 | 1261 |
| Abha | Base case | 32,476 | 5677 | 26,799 | 1286 | 5677 | 108 | 1394 |
| | SEC | | 12,356 | 20,120 | 966 | 12,356 | 235 | 1201 |
| | SBC-602 | | 13,602 | 18,874 | 906 | 13,602 | 258 | 1164 |

**Table 19.** Thermal insulation First-year saving for three cities.

| City | Thermal Insulation Requirements at (23 °C) | Saved Energy from Building Energy Need in the Base Case (kWh/Year) | First-Year Saving $ |
|---|---|---|---|
| Riyadh | SEC | 27,705 | 1330 |
| | SBC-602 | 43,283 | 2078 |
| Hail | SEC | 23,466 | 1126 |
| | SBC-602 | 34,519 | 1657 |
| Abha | SEC | 6679 | 321 |
| | SBC-602 | 7925 | 380 |

### 4.5. Financial Indicators

The LCC for all thermal insulation scenarios is equal to the capital cost of the system, which is $6536 and $19,381 for the thermal insulation as per SEC requirements and as per SBC-602 as shown in Table 9. Thermal insulation LCC is equal to capital cost since all the material is purchased only in the first year. There is no maintenance or operation cost during the project live, unlike the LCC of the PV system, including another cost besides the capital cost such as maintenance, inverter replacement, and the income from selling the system at the end of the project life. The LCC of unified PV is the same for all three cities with its three scenarios since the same size of the PV system is used. The LCC of the PV system was calculated by using Equation (3) equal to $21,806; the capital cost of the unified system was $13,707; total maintenance cost using the present value was $2184, the total replacement cost of the inverter was $7284, and the salvage value was $1368. All these costs were in today's money by using present worth for all future money of the unified PV system. The LCOE and LCSE are different from one scenario to another since it depends on produced or saved energy. The LCSE of thermal insulation requirements represents the cost to save one kWh. By applying the SEC requirements, the LCSE was 0.009, 0.011, and 0.039 $/kWh for Riyadh, Hail, and Abha, respectively. While fulfilling the thermal insulation conditions in the building code SBC-602, the LCSE will be 0.017, 0.023, and 0.098 $/kWh for Riyadh, Hail, and Abha, respectively. This LCSE for both SEC and SBC-602 is relatively high in Abha due to its cool weather, which leads to minimizing the A/C load. However, for Riyadh and Hail, some encouraging values were received in comparison to the PV LCOE, all the calculations of LCOE. The LCOE of the unified PV systems is between 0.026 and 0.031 $/kWh, which is propitious compared to other reported values. For example, from Table 3; 0.1 $/kWh in Kuwait [41], 0.036 $/kWh in Qassim [38], and 0.073–0.082 $/kWh Morocco [44]. However, the PV system's financial viability does not depend only on LCOE, the payback calculations, which take into account all the revenues and expenditures. The SPBT and NDPBT of thermal insulations are shown in Figure 10.

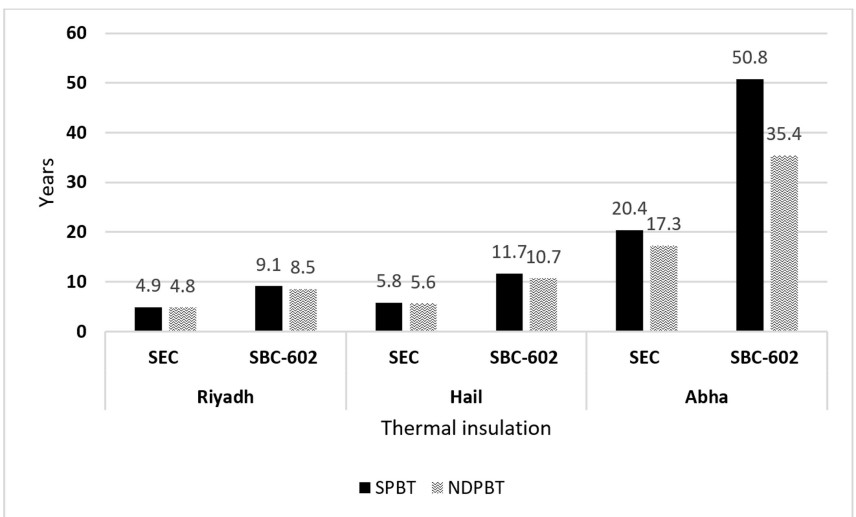

**Figure 10.** Simple payback time (SPBT) and not discounting payback time (NDPBT) of thermal insulation scenarios.

All the payback results were shown in Figure 11; it can be noted from the figure if the PV energy exceeds the building energy needs; the payback time will increase.

Kharseh et al. [53] estimated SPBT of fewer than two years by reducing the U-values of a wall from 1.76 to 0.57 W/m$^2$ K in residential buildings in Qatar. Esmaeil et al. [28] reported that SPBT value among 2.2–6.8 years was based on an electricity tariff by retrofitting residential building and Almasri et al. [29] between 10–15 years by applying the SBC-602 in the residential buildings in Qassim region, KSA. Note that there was a disagreement

between the current and previous values of SPBT, and that there is a discrepancy between the earlier values. The reasons could be climatic conditions, the price of the energy unit, the insulation materials, the labor cost, and most importantly, the conditions required to be achieved from the insulation process (the required heat transfer coefficient (U-value) and finally the date of the study. It is noted that to achieve the conditions of the thermal insulation code SBC-602, there is a good match between the SPBT results between Almasri et al. [29] and the current results for Riyadh and Hail.

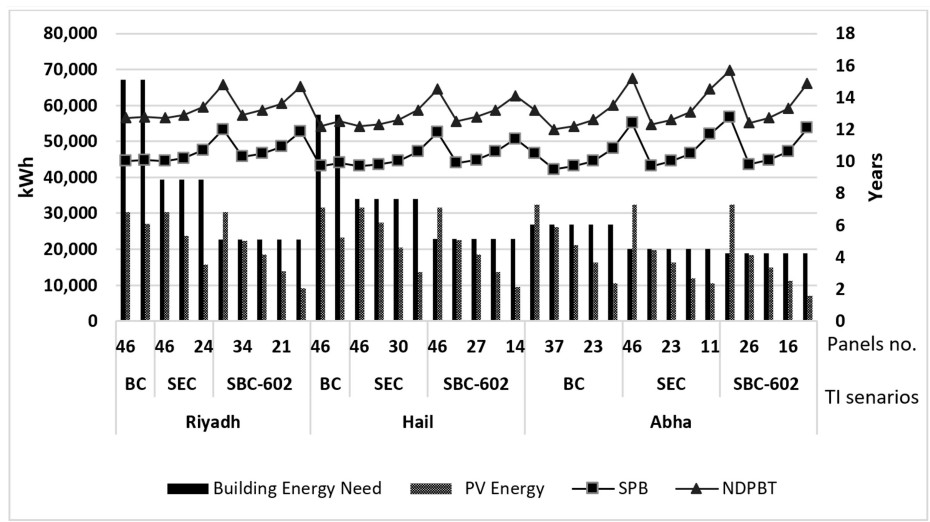

**Figure 11.** SPBT and NDPBT of all PV systems.

In general, the financial indicators of unified-sized PV systems were not encouraging compared to thermal insulation, significantly if the PV energy exceeds the building energy needs. It only happens when generated energy is more than they need due to the reason that ECRA tariff for electricity import is 0.048 $/kWh. In contrast, for export, the tariff is only 0.019 $/kWh. For example, the LCOE results for the PV system showing the best values were in Abha city. Nevertheless, its average payback of unified PV systems was the longest since the surplus energy is much more than the used energy. However, in Riyadh and Hail, the thermal insulation system's average payback is better than the PV. The last critical financial indicators were the net present value NPV. Table 20 illustrates the NPV of thermal insulation scenarios. NPV of thermal insulation in Abha results in minus values because of Abha's cold weather, so it is better to install a PV system. However, some of the PV scenarios indicate low profitable projects due to the oversize of the PV system. Therefore, it is essential to cover only the need for energy from the PV since there are no PV encouraging programs from the government. Therefore, when thermal insulation is applied, the energy need will decrease, so the size of the PV system also decreases to make the PV system a profitable project. Figure 12 summarizes the results from the RETScreen of PV systems NPV. Finally, the cumulative NPV between PV and thermal insulation is shown in Figure 13.

**Table 20.** NPV of thermal insulation scenarios for the three investigating cities.

| City | System Thermal Insulation | NPV $ |
|---|---|---|
| Riyadh | SEC | 18,953 |
| | SBC-602 | 20,443 |
| Hail | SEC | 15,044 |
| | SBC-602 | 12,375 |
| Abha | SEC | −384.1 |
| | SBC-602 | −12,098 |

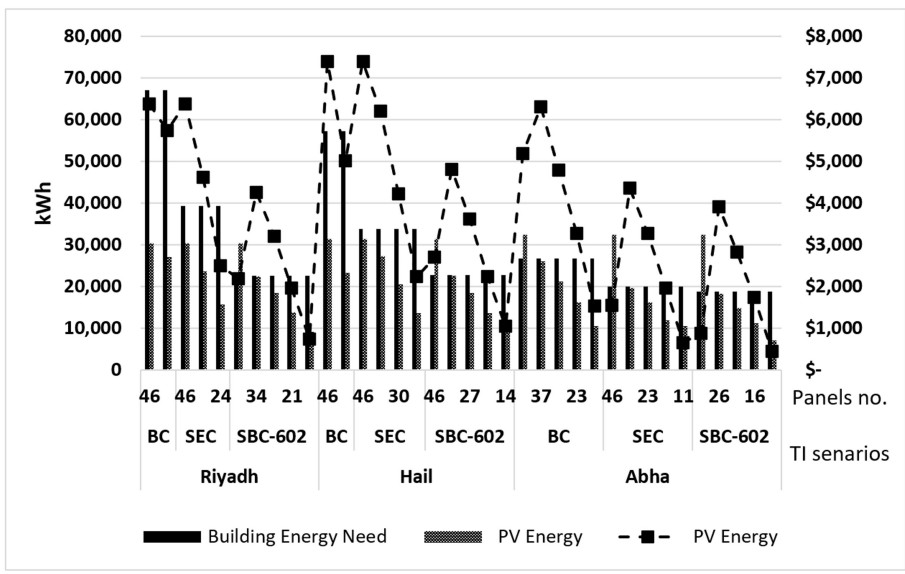

**Figure 12.** NPV for all PV systems.

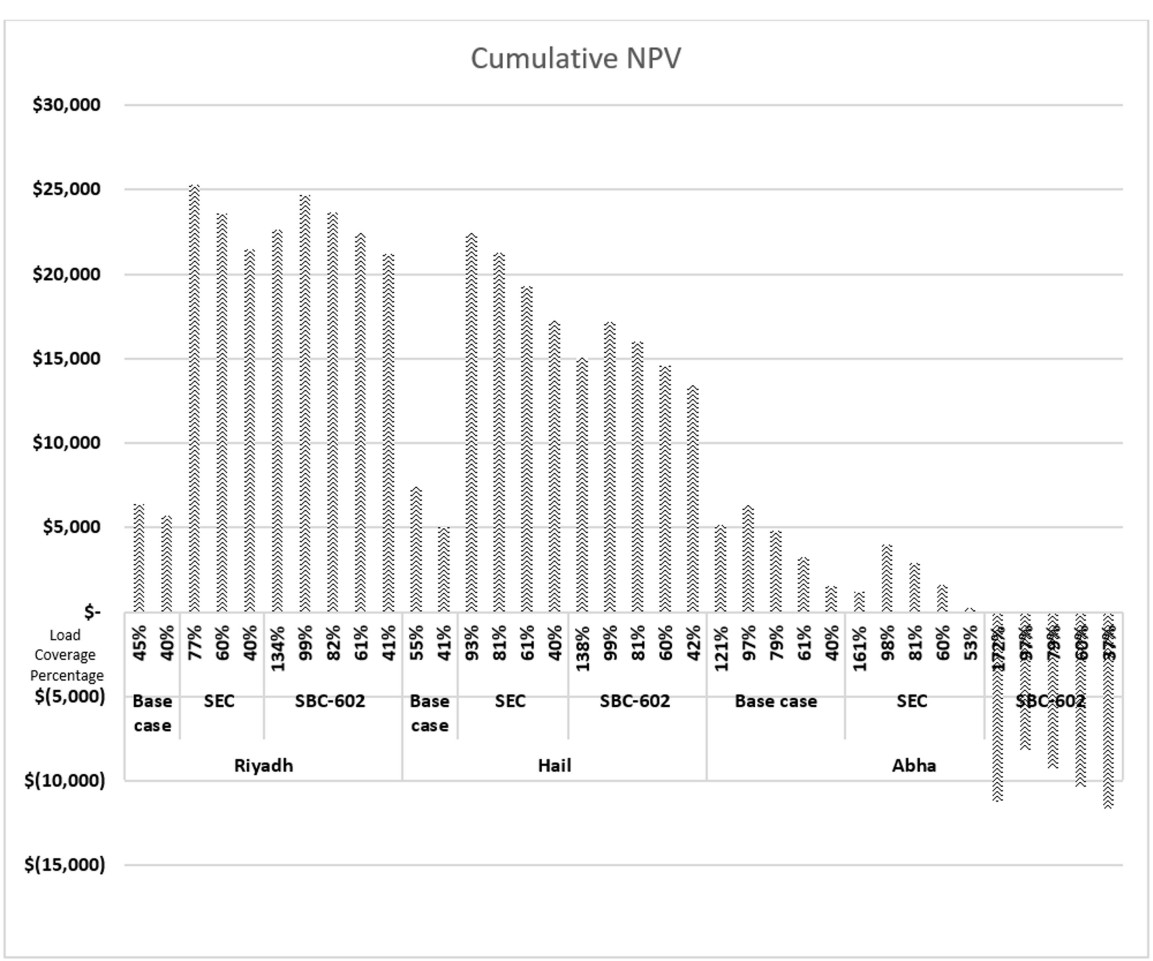

**Figure 13.** Cumulative NPV between PV and thermal insulation.

### 4.6. Environmental Analysis

The environmental impact caused by applying thermal insulation and using the PV system in a residential building will be assessed. Being a full signatory of the Paris Agreement (3 November 2016) [54], Montreal Protocol (1 March 1993) [55], and Kyoto

Protocol (31 January 2005) [56], it is an international obligation that KSA should practice environmentally benign refrigerants in heating and cooling applications and implement practical measures to abbreviated GHG emissions from the country. Hence, incorporated efforts may be made to sustain the planet earth's crest temperature below 1.5 °C by 2030. This situation has given way to the naturally occurring refrigerants to replace the chlorofluorocarbon (CFCs) and hydrochlorofluorocarbon (HCFCs) [57]. The Paris agreement decided to control the global rise in temperature by not allowing it to go beyond 2 °C by the end of 2030 and advance more efforts to limit this increase to 1.5° [54].

Applying thermal insulation and using the PV system will reduce GHG emissions in the residential buildings in KSA. As per SEC requirements, thermal insulation will have an annual saving of electricity consumption of about 27,705; 23,466; and 6679 kWh for Riyadh, Hail, and Abha, respectively, and from thermal insulation as per SBC-602, it will produce an annual saving of electricity consumption of about 44,441; 34,519; and 7952 kWh for Riyadh, Hail, and Abha, respectively. These savings of electricity consumption of these buildings will cause annual GHG emissions reduction, as shown in Table 21. The GHG analysis was done according to the United States Environmental Protection Agency [56]. The energy mix of 2019 revealed that around 57% of electricity was generated by gas and 43% from oil [11]. The RETScreen obtained the results with a conversion factor of 0.866 and 0.67 $tCO_2$/MWh for oil and gas, respectively [50]. The yearly GHG emissions reductions by unified PV systems were 22.9, 23.8, and 24.5 $tCO_2$ for Riyadh, Hail, and Abha, respectively, as shown in Table 21.

**Table 21.** Yearly GHG emissions reduction by applying thermal insulation and a Unified PV system.

| City | GHG Emissions Reduction ($tCO_2$) | | |
|---|---|---|---|
| | SEC | SBC-602 | Unified PV Systems |
| Riyadh | 19.6 | 31.4 | 22.9 |
| Hail | 16.6 | 24.4 | 23.8 |
| Abha | 4.7 | 5.6 | 24.5 |

In Europe, countries are pushing to maintain their European Union targets for sustainable development [58]. Solar energy, especially PV, is being studied in the world's major metropolitan cities of China [59] (Beijing) and Italy [60] (Rome, Naples, and Milan), as investigated by different researchers.

## 5. Conclusions and Recommendations

This paper investigates the influence of SBC-602 on the solar PV system cost in residential buildings in the KSA. The cooling demand in KSA is very high, which in turn directly influences GHG emissions. The result indicates that decreasing the external wall's U-value is the most encouraging way to reduce the building electrical consumption in the residential sector. The results for energy consumption are given below:

- The annual electrical energy consumption of the building base case in Riyadh city was the highest 67,095 kWh, while for the Hail, it was 57,373 kWh and Abha 26,799 kWh.
- In Riyadh city, 69% of the total energy was used for cooling and heating for the basic case-building, and by applying the SBC-602 requirement, it will be only 19%.

After the energy analysis was done, an economic study on thermal insulation options was done for the three cities. The SPBT of SEC requirements were 4.9, 5.8, and 20.4 years for Riyadh, Hail, and Abha, respectively, and SPBT of SBC-602 conditions are 9.1, 11.7, and 50.8 years for Riyadh, Hail, and Abha, respectively.

The techno-economic feasibility of roof-mounted on-grid solar PV system was performed, and the proposed PV system showed high energy productivity, achieved up to 30,348; 31,485; and 32,476 kWh/year for Riyadh, Hail, and Abha, respectively. The performance indicators were good, and so was the annual YF for Riyadh, Hail, and Abha. The key

findings from the effect of thermal insulations on the economic feasibility of grid-connected PV energy system:

- The NPV of SEC thermal insulation requirement in Riyadh city was the highest $18,953, and Hail was $15,044, while for Abha, it was negative −$384.1.
- The NPV of SBC-602 thermal insulation requirement in Riyadh city was the highest $20,443, and Hail was $12,375, while for Abha, it was negative −$12,098.
- The cumulative NPV of SEC thermal insulation requirement with unified PV was $25,334, $22,437, and $1177 for Riyadh, Hail, and Abha, respectively.
- The cumulative NPV of SBC-602 thermal insulation requirement with unified PV was $22,643, $15,077, and −$11,214 for Riyadh, Hail, and Abha, respectively.
- The best NPV results were when combined SEC thermal insulation requirement with a PV system covering 75–100% of the required load.
- In Riyadh, the capital cost of the PV system covering 80% of the base caseload was around $24,260. When the SEC thermal insulation requirement was applied, it could decrease up to 170%, and when the SBC-602 thermal insulation requirement was applied, it could decrease up to 296%.
- In Hail, the capital cost of the PV system covering 80% of the base caseload was $19,973; when the SEC thermal insulation requirement was applied, it could decrease up to 169%, and when SBC-602 thermal insulation requirement applied, it could decrease up to 251%.
- In Abha, the capital cost of the PV system covering 80% of the base caseload cost around $9048; when the SEC thermal insulation requirement applied, it could decrease up to 133%, and when SBC-602 thermal insulation requirement applied, it could decrease up to 142%.

Through insulation in homes, energy can be conserved, and then automatically, the electricity bill will be reduced. With reduced energy demand, a low-capacity solar PV system can also efficiently fulfill the energy demands economically, especially since the prices of PV systems are falling continuously, but the government must support it.

The following are some recommendations drawn after concluding the results.

- The instructions of the SBC-602 Code must be applied entirely in a new building in zones 1 and 2 and granting owners of old buildings interest-free loans to implement the required conditions in zone 1 and 2.
- Reduce the required conditions of the SBC-602 Code in the climatic zone 3.
- Implement government support programs to perform more research and development to take advantage of the available RE for water heating and solar thermal and electrical cooling in KSA.
- Carry out qualitative and quantitative research to study human behavior towards energy use in residential areas.

Due to favorable solar insulation and high-temperature climate in KSA, the use of solar thermal systems, like a flat plate evacuated glass tube and concentrated collectors to capture the maximum solar energy, is an imminent future research option. The solar thermal heating and cooling systems are of particular interest in KSA's hot climate, and there is a need for promising research in these areas for KSA.

The primary limitations for applying the SBC-602 Code and simultaneously using solar PV only face low FIT of only 0.019 $/kWh, which is significantly less than the cost taken by the utility of electricity supplied by them. Increasing this value will allow more investors and residents to go for solar PV technology in the KSA.

**Author Contributions:** R.A.A., investigation, original draft, methodology, and review. A.A.A., data curation, original draft, investigation, formal analysis, and editing. S.D. analysis, writing, and editing. All authors have read and agreed to the published version of the manuscript.

**Funding:** This research received no external funding.

**Institutional Review Board Statement:** Not applicable.

**Informed Consent Statement:** Not applicable.

**Data Availability Statement:** Not applicable.

**Acknowledgments:** Researchers would like to thank the Deanship of Scientific Research, Qassim University for funding the publication of this project.

**Conflicts of Interest:** The authors declare no conflict of interest.

## Abbreviations

| | |
|---|---|
| AC | Alternating Current |
| A/C | Air Conditioning |
| BOS | Balance of system |
| CDD | Cooling Degree-Days |
| CF | Capacity factor |
| COP | Coefficient of Performance |
| ECRA | Electricity and Co-Generation Regulatory Authority |
| EIFS | Exterior insulation finishing system |
| EE | Energy Efficiency |
| EUI | Energy Used Intensity |
| FIT | Feed-in tariff |
| GCC | Gulf Cooperation Council |
| GHG | Greenhouse Gases |
| GHI | Global Horizontal Irradiance |
| GRC | Ground Coverage Ratio |
| HCB | Hollow Concrete Blocks |
| HDD | Heating Degree-Days |
| IEA | International Energy Agency |
| KSA | Kingdom of Saudi Arabia |
| LCC | Life cycle cost |
| LCOE | Levelized cost of energy |
| LCSE | Levelized cost of saved energy |
| MPPT | Maximum Power Point Tracking |
| NDPBT | |
| NPV | Net present value |
| NZEB | net-zero energy building |
| PV | Photovoltaic |
| RE | Renewable Energy |
| SBC | Saudi Building Code |
| SEC | Saudi Electricity company |
| SPBT | Simple payback time |
| SPD | Surge protection device |
| STC | Standard test conditions |
| U-value | Heat transfer coefficient |
| V | Voltage |
| W | Watt |
| Wh | Watthour |
| YF | Yield factor |

## Appendix A

**Table A1.** Statistical weather report of Riyadh [9].

| Month | Air Temperature (°C) | Relative Humidity (%) | Precipitation (mm) | Daily Solar Radiation Horizontal (kWh/m²/d) | Atmospheric Pressure (kPa) | Wind Speed * (m/s) | Earth Temperature (°C) |
|---|---|---|---|---|---|---|---|
| January | 14.0 | 46.2 | 18.75 | 3.50 | 94.7 | 0.5 | 15.9 |
| February | 16.4 | 36.4 | 8.70 | 4.60 | 94.5 | 0.6 | 18.9 |
| March | 21.1 | 33.7 | 16.86 | 5.10 | 94.2 | 0.6 | 23.4 |
| April | 25.7 | 28.5 | 16.87 | 5.50 | 93.9 | 0.5 | 29.8 |
| May | 31.5 | 17.1 | 1.21 | 5.60 | 93.6 | 0.5 | 35.8 |
| June | 34.2 | 10.4 | 0.08 | 6.10 | 93.1 | 0.6 | 38.1 |
| July | 35.0 | 9.9 | 0.04 | 6.10 | 92.8 | 0.6 | 39.8 |
| August | 35.1 | 11.9 | 0.22 | 5.90 | 93.0 | 0.6 | 39.5 |
| September | 31.9 | 13.5 | 0.27 | 5.70 | 93.5 | 0.4 | 36.1 |
| October | 26.8 | 20.3 | 1.52 | 5.30 | 94.1 | 0.3 | 30.3 |
| November | 20.7 | 36.2 | 14.38 | 4.50 | 94.5 | 0.3 | 23.7 |
| December | 15.4 | 47.5 | 14.15 | 3.60 | 94.7 | 0.3 | 18.0 |
| Annual | 25.7 | 25.9 | 93.05 | 5.13 | 93.9 | 0.5 | 29.2 |
| Source | Ground | Ground | NASA | Ground | Ground | Ground | NASA |

* Measured at 10 m.

**Table A2.** Statistical weather report of Hail [9].

| Month | Air Temperature (°C) | Relative Humidity (%) | Precipitation (mm) | Daily Solar Radiation Horizontal (kWh/m²/d) | Atmospheric Pressure (kPa) | Wind Speed * (m/s) | Earth Temperature (°C) |
|---|---|---|---|---|---|---|---|
| January | 10.4 | 53.3 | 18.28 | 3.46 | 90.8 | 3.2 | 11.8 |
| February | 12.4 | 43.3 | 8.92 | 4.45 | 90.7 | 3.4 | 15.0 |
| March | 16.3 | 38.6 | 11.82 | 5.23 | 90.5 | 3.7 | 20.0 |
| April | 22.1 | 33.4 | 6.12 | 6.19 | 90.5 | 3.8 | 26.7 |
| May | 27.4 | 24.0 | 5.35 | 6.42 | 90.4 | 3.7 | 32.9 |
| June | 31.2 | 16.0 | 0.21 | 6.79 | 90.2 | 3.3 | 36.1 |
| July | 32.5 | 15.9 | 0.36 | 6.60 | 89.9 | 3.2 | 38.6 |
| August | 32.8 | 17.0 | 0.46 | 6.22 | 90.0 | 2.9 | 38.8 |
| September | 30.2 | 18.2 | 0.22 | 5.63 | 90.4 | 2.7 | 35.4 |
| October | 24.5 | 27.3 | 5.64 | 4.82 | 90.7 | 3.0 | 28.5 |
| November | 17.0 | 46.9 | 14.21 | 3.66 | 90.9 | 2.9 | 19.7 |
| December | 12.0 | 53.1 | 13.25 | 3.36 | 90.9 | 2.9 | 13.6 |
| Annual | 22.5 | 32.2 | 84.84 | 5.24 | 90.5 | 3.2 | 26.5 |
| Source | Ground | Ground | NASA | Ground | Ground | Ground | NASA |

* Measured at 10 m.

**Table A3.** Statistical weather report of Abha [9].

| Month | Air Temperature (°C) | Relative Humidity (%) | Precipitation (mm) | Daily Solar Radiation Horizontal (kWh/m²/d) | Atmospheric Pressure (kPa) | Wind Speed * (m/s) | Earth Temperature (°C) |
|---|---|---|---|---|---|---|---|
| January | 13.2 | 70.2 | 10.78 | 4.74 | 79.8 | 3.8 | 20.7 |
| February | 14.6 | 67.8 | 1.00 | 4.60 | 79.8 | 4.4 | 22.9 |
| March | 16.5 | 64.4 | 16.34 | 5.37 | 79.7 | 4.2 | 25.2 |
| April | 18.3 | 60.8 | 27.23 | 5.62 | 79.7 | 3.3 | 28.0 |
| May | 21.1 | 50.6 | 26.47 | 5.89 | 79.8 | 2.7 | 32.3 |
| June | 23.3 | 39.1 | 7.59 | 6.01 | 79.6 | 2.7 | 33.3 |
| July | 23.2 | 44.4 | 4.63 | 5.52 | 79.5 | 3.0 | 30.0 |
| August | 22.6 | 51.7 | 13.25 | 5.30 | 79.6 | 2.8 | 28.4 |
| September | 21.9 | 38.9 | 6.74 | 5.73 | 79.7 | 2.9 | 31.0 |
| October | 18.5 | 43.6 | 15.00 | 6.02 | 79.9 | 2.5 | 28.8 |
| November | 15.6 | 61.0 | 16.30 | 5.50 | 79.9 | 2.4 | 24.3 |
| December | 13.8 | 67.1 | 13.74 | 4.81 | 79.9 | 3.0 | 21.4 |
| Annual | 18.6 | 54.9 | 159.07 | 5.43 | 79.7 | 3.1 | 27.2 |
| Source | Ground | Ground | NASA | Ground | Ground | Ground | NASA |

* Measured at 10 m.

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
