# Peer review of "Investigating the Impact of Integration the Saudi Code of Energy Conservation with the Solar PV Systems in Residential Buildings"

_sustainability, doi:10.3390/su13063384_

Round 1

Reviewer 1 Report

The energy for cooling and heating in the building are consumed significantly in some areas. How to reduce the energy economically is very important. Radwan et al. investigate the impact of integration the Saudi code of energy conservation with the solar PV systems in residential buildings in KSA. The cooling demand in KSA is very high, which in turn directly influences GHG emissions. The result indicates that decreasing the external wall's U-value is the most encouraging way for the overall reduction of the building electrical consumption in the residential sector. The economic feasibility was determined using the life cycle analysis method and the payback time method for both thermal insulation and PV system.  The environmental impact was listed from the energy saved and green energy to be a more sustainable building. The authors present a comprehensive work systematically.

Author Response

Many thanks for your positive feedback. The paper has been checked thoroughly to correct the typing mistakes.

Thank you very much.

Reviewer 2 Report

The main goal of the presented work was to investigate the energy consumption usage dependency on thermal insulation and the size of photovoltaic systems for residential buildings located in three different places in Saudi Arabia. Economic and environmental aspects of the particular type of building’s insulation as well as a photovoltaic system were taken into account. The analysis was carried out mostly based on simulation results. The main concerns were listed below.

  • Table 15. Yield factors presented in Tab. 15 should be verified. According to the eq. (1) and PV energy production data shown in Tab. 14, the Annual Yield Factors should be between 1649 kWh/kWp and 1765 kWh/kWp. Verify and correct.
  • Table 16 presents monthly energy production in kWh. The capacity of the PV system should be defined for this kind of data.
  • lines 460 – 462: This part of the text should be in the Introduction section.
  • Table 17: It is not clear how the ‘average unified PV energy’ was calculated. Add the explanation.
  • In many tables, the same data (information) can be found. For example, 67095 kWh can be found in Tabs. 14, 17, 18. Try to avoid repeating the same data in different tables.
  • All variables and numbers (8760, 0.18, 0.07, etc.) in the equations (1-10) must be explained in the text.
  • Equations (6), (7), (8), (9): use USD currency instead of local SAR currency (as it is in the whole paper).
  • Something is missing in equation (7). It is impossible to verify the calculations without a proper equation. Verify and correct.

Reviewer 3 Report

The article deals with an insightful topic as well as quite studied. This is why the authors must stress the novelty of their work.

Key highlights are not in the template.

Literature background can start from high-level perspective down to the local context. See recent and highly cited studies in Energies, Frontiers in Energy Research, Energy Procedia and others. Approach the topic from National via Regional to Urban level. As example for each source find https://doi.org/10.3390/en13020417 https://doi.org/10.3389/fenrg.2019.00155 https://doi.org/10.1016/j.egypro.2016.11.136

Check carefully Figures' style as well as their composition. For instance in Figure 12, x axis is not clear. Are years or options or locations changing?

Check English spell and stress in the conclusions, the identified research gap, how the authors addressed it and what potential and limitations to their approach.

Round 2

Reviewer 2 Report

Authors have addressed all my comments. Some minor recommendations were listed below.

  • 78: values 4400 Wh/m2 or 7300 Wh/m2 are rather irradiation than irradiance (unit for irradiance is W/m2).
  • verify the distance between the value and the unit in the whole text (308, 458 for example).
  • 343: delete :(
  • 413-414: there is no need to use Bold style font here.
  • Tab. 18: verify and correct the values of money-saving ($) in FIT of tab. 18. In my opnion the values should be slighty different (124 instead of 122 for example etc.).
  • 599: use energy instead of 'Energy'.
  • 647: change the font color to black.

Reviewer 3 Report

The article has been improved as suggested.

Two remarks remain to be solved:

the insertion of the considered references in the text since they are at the moment just listed at the end of the paper

a check of English spell.
